# Flooding Tolerance in Sweet Potato (*Ipomoea batatas* (L.) Lam) Is Mediated by Reactive Oxygen Species and Nitric Oxide

**DOI:** 10.3390/antiox11050878

**Published:** 2022-04-29

**Authors:** Sul-U Park, Chan-Ju Lee, Sung-Chul Park, Ki Jung Nam, Kang-Lok Lee, Sang-Soo Kwak, Ho Soo Kim, Yun-Hee Kim

**Affiliations:** 1Plant Systems Engineering Research Center, Korea Research Institute of Bioscience and Biotechnology (KRIBB), 125 Gwahak-ro, Daejeon 34141, Korea; sulu0849@kribb.re.kr (S.-U.P.); moda22@kribb.re.kr (C.-J.L.); sskwak@kribb.re.kr (S.-S.K.); 2Department of Environmental Biotechnology, KRIBB School of Biotechnology, University of Science and Technology (UST), 217 Gajeong-ro, Daejeon 34113, Korea; 3Biological Resource Center, Korea Research Institute of Bioscience and Biotechnology (KRIBB), Jeongeup 56212, Korea; heypsc@kribb.re.kr; 4Department of Biology Education, IALS, Gyeongsang National University, Jinju 52828, Korea; prin225@gnu.ac.kr (K.J.N.); leekl@gnu.ac.kr (K.-L.L.)

**Keywords:** ethylene response factor, flooding stress, metallothionein, monodehydroascorbic acid reductase, resistant cultivar, respiratory burst oxidase, sensitive cultivar

## Abstract

Flooding is harmful to almost all higher plants, including crop species. Most cultivars of the root crop sweet potato are able to tolerate environmental stresses such as drought, high temperature, and high salinity. They are, however, relatively sensitive to flooding stress, which greatly reduces yield and commercial value. Previous transcriptomic analysis of flood-sensitive and flood-resistant sweet potato cultivars identified genes that were likely to contribute to protection against flooding stress, including genes related to ethylene (ET), reactive oxygen species (ROS), and nitric oxide (NO) metabolism. Although each sweet potato cultivar can be classified as either tolerant or sensitive to flooding stress, the molecular mechanisms of flooding resistance in ET, ROS, and NO regulation-mediated responses have not yet been reported. Therefore, this study characterized the regulation of ET, ROS, and NO metabolism in two sweet potato cultivars—one flood-tolerant cultivar and one flood-sensitive cultivar—under early flooding treatment conditions. The expression of *ERFVII* genes, which are involved in low oxygen signaling, was upregulated in leaves during flooding stress treatments. In addition, levels of respiratory burst oxidase homologs and metallothionein-mediated ROS scavenging were greatly increased in the early stage of flooding in the flood-tolerant sweet potato cultivar compared with the flood-sensitive cultivar. The expression of genes involved in NO biosynthesis and scavenging was also upregulated in the tolerant cultivar. Finally, NO scavenging-related *MDHAR* expressions and enzymatic activity were higher in the flood-tolerant cultivar than in the flood-sensitive cultivar. These results indicate that, in sweet potato, genes involved in ET, ROS, and NO regulation play an important part in response mechanisms against flooding stress.

## 1. Introduction

Flooding is a major environmental stress that causes loss of crop yield in various agricultural settings. The global risk of flooding has increased dramatically in recent decades due to climate change [1]. Over the past 50 years, increasingly frequent and severe flood events have negatively impacted the lifespan of terrestrial plants. When flooding occurs, gas exchange between the plant and its environment is severely restricted, causing several internal changes in the plant. Plant organs that are submerged in water lack O_2_ and/or CO_2_ and may accumulate high levels of the gaseous hormone ethylene (ET). In addition, there are changes in the concentrations of oxygen-derived free radicals, reactive oxygen species (ROS), and nitric oxide (NO) [2]. The production, kinetics, concentration, and precise balance of these substances in flooded cells depend on the plant organ and the type of flooding. Some crop species can withstand soil inundation for only a few hours, while other flood-tolerant crops can cope with partial or complete flooding for days or months [2].

Flood-tolerant crops have anatomical, morphological, or metabolic adaptations. One of the responses to flooding is the induction of ET [3]. ET production is enhanced in crops such as rice and barley. Only flood-tolerant rice varieties, however, can withstand prolonged soil submersion or submergence phases [2]. ET is usually produced in a two-step reaction involving 1-aminocyclopropane-1 carboxylate (ACC) synthetase and ACC oxidase. ET is not only produced in response to flooding stresses but is also a modulator of stress-related morphological responses during plant growth [4,5]. In *Arabidopsis thaliana*, ET is sensed by ET receptors (ETRs), which trigger a downstream signaling cascade of transcription factors (TFs) including *APETALA2/ETHYLENE RESPONSE FACTOR* (*AP2/ERF*) [6,7].

ROS are among the key molecules mediating plant responses to flood stress [8]. Apoplastic ROS are usually generated via the respiratory burst oxidase homolog (RBOH) protein located in the plasma membrane. RBOH is a homolog of the mammalian NADPH oxidase subunit gp91phox that produces a superoxide anion (O_2_^−^) [9]. The short-lived and highly reactive toxic chemical O_2_^−^ is converted to the nonradical hydrogen peroxide (H_2_O_2_), either spontaneously or by catalysis by superoxide dismutase (SOD) [10]. ROS might also be formed by mitochondria and/or the photosynthetic electron transport chain [8,10]. Nonenzymatic ROS-scavenging proteins, including cysteine-rich metallothionein (MT) and antioxidant enzymes such as catalase and ascorbate-glutathione cycle-related enzymes, are essential for ROS homeostasis. During flooding stress, the ROS balance of cells is disturbed due to either enhanced ROS production or a decrease in ROS scavenging capacity [8].

NO is a short-lived, highly reactive molecule that regulates several plant developmental and stress responses [11,12]. The main method of NO production is through the enzymatic and nonenzymatic reduction of nitrites [13,14]. Nitrite production is highly dependent on nitrate reductase (NR) activity. The dependence of NO production on nitrite availability means that NR plays the major role in NO production [14,15]. There are limited and contrasting data on the temporal and spatial dynamics of NO in flooded plants [15,16,17]. NO emissions from unsubmerged atmospheric plant tissues decreased in Arabidopsis and cotton under waterlogging, whereas NO emissions increased in three submerged deciduous tree species [15,16,17]. In addition, cellular and exogenous NO levels increased in various plant species during hypoxia [13,18,19]. During hypoxic conditions, this NO burst is thought to reflect an increase in NO production due to increased NR activity and nitrite accumulation. The spatiotemporal dynamics of NO and its effects may vary, therefore, depending on the flood conditions. Studying the changes in ET, ROS, and NO levels is thus important for fully understanding the mechanisms of plant flooding responses and tolerance.

We previously studied 33 sweet potato cultivars, using phenotypic and biochemical characteristics to identify flood-tolerant cultivars [20]. In addition, we recently used comparative transcriptome profiling to compare a flood-tolerant sweet potato cultivar, Yeonjami (YJM), with a flood-sensitive cultivar, Jeonmi (JM) [21]. The expression levels of several candidate genes thought to be involved in flooding tolerance correlated with the comparative transcriptomic data. However, although each sweet potato cultivar can be classified as either tolerant or sensitive to flooding stress, the molecular mechanisms of flooding resistance in ET, ROS, and NO regulation-mediated responses have not yet been reported. In this study, therefore, we conducted a transcriptome-based expression analysis of genes involved in ET, ROS, and NO regulation in two sweet potato cultivars during flooding stress.

## 2. Materials and Methods

### 2.1. Plant Materials

This study involved two previously characterized cultivars of sweet potato (*Ipomoea batatas* (L.) Lam), namely, Yeonjami (YJM) and Jeonmi (JM). YJM is highly tolerant to flooding stress, whereas JM is highly sensitive [20,21]. Both cultivars were obtained from the Bioenergy Crop Research Center, National Crop Research Institute (RDA, Muan, Jeonnam, Korea). Sweet potato plants were cultivated and subjected to flooding stress as described by Park et al. [21]. Plants were grown in a growth chamber at 25 °C under a 16 h/8 h light/dark photocycle and 60% relative humidity. Experimental plants were grown for 2 weeks under normal conditions and then subjected to flooding stress. To induce flooding stress, water was added to the pots until approximately 65% of the aboveground tissue was submerged. Leaves were sampled 0 (control), 0.5, and 3 days after the beginning of flooding treatment. For leaf samples used in the experiment, the 3rd and 4th leaves from the top were used in four plants. The harvested leaves were above the water surface. Leaf samples were frozen immediately in liquid nitrogen and stored at −70 °C until further analysis. 

### 2.2. Analysis of H_2_O_2_ Levels

The H_2_O_2_ content of sweet potato leaves was assessed using the xylenol orange method [22]. A total of 0.1 g of leaf tissue was ground and homogenized in a solution of 50 mM potassium phosphate buffer (pH 6.5) and 1 mM hydroxylamine. The homogenate was centrifuged at 12,000× *g* for 15 min at 4 °C. A total of 100 μL of the supernatant was added to a reaction mixture containing ferric ammonium sulfate (FeNH_4_[SO_4_]), 25 mM sulphuric acid, 250 μM xylenol orange, and 100 mM sorbitol. After 30 min in the dark at room temperature for incubation, the absorbance of the samples was determined at 560 nm. H_2_O_2_ measurements were expressed as relative values by control and treatment.

### 2.3. Analysis of Nitrite Content

The nitrite (NO_2_^−^) content was analyzed using a Promega Griess Reagent System assay kit (Promega, USA). Sweet potato leaves were homogenized on ice using a pestle and mortar in 25 mM HEPES buffer (pH 7.7), incubated for 20 min at 4 °C, and centrifuged for 10 min at 12,000 rpm. The NO_2_^−^ concentration was determined using the Griess Reagent System, according to the manufacturer’s instructions. Briefly, 40 µL of each sample was added to 50 µL sulfanilamide solution in a 96-well microtiter plate. The microtiter plate was covered and incubated at room temperature for 10 min before 40 µL NED (N-(1-naphthyl) ethylenediamine) solution was added to each well. The plate was then incubated at room temperature for a further 10 min. The absorbance of nitrite was read at 540 nm using a plate reader.

### 2.4. RNA Isolation and Analysis of Gene Expression

Total RNA was isolated from leaves using TRIzol reagent (Invitrogen, Carlsbad, CA, USA) and treated with RNase-free DNase I to avoid contamination with genomic DNA. Quantitative real-time PCR analysis was performed using a Bio-Rad CFX96 thermal cycler (Bio-Rad) with EvaGreen fluorescent dye, according to the manufacturer’s instructions. Linear data were normalized to the mean threshold cycle (Ct) of the reference gene *ADPRIBOSYLATION*
*FACTOR* (*ARF*) [23]. Gene-specific PCR primers are listed in Appendix A. 

### 2.5. Monodehydroascorbate Reductase Assay

Monodehydroascorbate reductase (MDHAR) activity was measured in sweet potato leaves according to the methodology of Truffault et al. [24]. The MDHAR enzymatic activity assay is based on NADH oxidation. Extractions were performed on ground sweet potato leaf powder in 50 mM Tris-HCl at pH 7.8. The soluble extract was mixed with 1 mM ascorbate, 0.2 mM NADH, and ascorbate oxidase to give a linear production of the monodehydroascorbate (MDHA) radical. Measurements were performed in triplicate at 340 nm. 

### 2.6. Statistical Analyses

Data were analyzed by two-way analysis of variance (ANOVA). The subsequent multiple comparisons were examined using the least significant difference (LSD) multiple range test. All statistical analyses were performed using Statistical Package for the Social Sciences software (SPSS 12). Statistical significance levels were set at *p* < 0.05.

## 3. Results

### 3.1. Expression of ERFVII during Early Flooding Treatment

To analyze the responses of sweet potato plants to flood stress, the tolerance of whole plants grown in soil was measured during early flooding treatment. Approximately 65% of the aboveground tissue of 2-month-old plants was submerged by adding excess water to the pot in a growth chamber at 25 °C under a 16 h/8 h light/dark photocycle. When the susceptible cultivar JM and the resistant cultivar YJM were exposed to flooding for 0, 0.5, and 3 days, the above atmospheric leaves of the JM plants showed slight wilting and curling, whereas the leaves of the YJM plants were less damaged (Figure 1A).

ET biosynthesis and signaling pathways are usually activated during flood treatment in plants [25]. TFs belonging to the *ERFVII* group, including *ERF71/HER2*, *ERF72/RAP2.3*, *ERF74/RAP2.12*, and *ERF75/RAP2.2*, are important representatives of genes that function in the ET signaling pathway activated during flood treatment at the transcriptional and post-translational levels [26,27]. Although expression analysis at the post-translational level is more important in the *ERFVIIs* [27], in this study, the expression of *ERFVII* group genes was investigated during early flood treatment at the transcriptional level (Figure 1B). The expression patterns of *ERF72/RAP2.3* (g6122), *ERF74/RAP2.12* (g55154), and *ERF74/RAP2.12* (g60549) were investigated after flood treatment using our previous transcriptomic data [21]. *ERF72/RAP2.3* was strongly induced from 0.5 d after flooding treatment in the flood-tolerant cultivar, YJM, whereas its expression decreased in the flood-sensitive JM. In contrast, *ERF74/RAP2.12* showed a weak increase in expression level in YJM, and increased expression after 3 d of flooding treatment. In JM, there was no clear pattern of changes in expression under flooding conditions, as levels increased or decreased at different times.

### 3.2. The Responses of ROS during Early Flooding Treatment

The levels of free radicals and ROS in plants usually increase during flood treatment [25]. Increases in ET biosynthesis and signaling during flood treatment modulate the amount of ROS and free radical signaling. Therefore, changes in H_2_O_2_, a representative ROS, were investigated during the early flood treatment process (Figure 2A). On day 3 of flood treatment, the H_2_O_2_ content of the flooding-tolerant cultivar, YJM, was 38% lower than that of the flooding-sensitive cultivar, JM. 

We also investigated the expression levels of *RBOH*, a gene which is involved in early ROS production, during flood treatment (Figure 2B). Expression levels of *RBOHA* (g19623), *D* (g51882), and *E* (g55049) were higher in the resistant cultivar YJM than in the susceptible JM under normal untreated conditions; expression increased to higher levels in YJM than in JM within 0.5 days of flood treatment. On the other hand, *RBOHC* (g19835) showed similar changes in expression levels in both cultivars under the normal control and the early flooding treatment. After 3 days of flood treatment, the expression levels of all the *RBOH* genes measured in the experiment were higher in JM than in YJM, due to decreases in expression in the resistant YJM cultivar between 0.5 and 3 days of flood treatment. 

Next, we measured the expression of *METALLOTHIONEIN* (*MT*) genes, which encode proteins that remove reactive oxygen species during flood treatment (Figure 2C). Expression levels of all the *MT* genes were higher in the flood-resistant YJM than in the flood-sensitive JM under normal untreated conditions. After 0.5 days of flood treatment, expression levels of *MT* genes remained higher in YJM than in JM; in particular, *MT2* (g58960 and g4909) showed a greater increase in expression in YJM than in JM. In contrast, *MT3* (g64044) showed similar expression levels in both cultivars under both control conditions and early flooding treatment. By the third day of flood treatment, however, the expression levels of *MT1* and *MT2* decreased in YJM and increased in JM.

### 3.3. The Responses of NO during Early Flooding Treatment

An increase in NO, together with rises in ET and ROS levels, is seen in plants at the initial stage of flood treatment; this increase in NO controls the increase in ET during flood treatment [26]. Plants usually regulate endogenous NO levels via the control of biosynthesis and scavenging. The main source of NO production is via the enzymatic and nonenzymatic reduction of nitrite (NO_2_^−^) [13,14]. NO_2_^−^ production is highly dependent on nitrate reductase (NR). The dependency of NO production on nitrite availability makes NR the major player in NO production [14,15]. The NO generation during hypoxia is also thought to result from enhanced NR activity and NO_2_^−^ accumulation, providing a substrate for NO production [13,19].

We therefore tested the changes in nitrite (NO_2_^−^) levels in sweet potato plants during early flood treatment. In contrast to the changes in H_2_O_2_ levels (Figure 2A), the increase in NO_2_^−^ was 31% higher in YJM than in JM (Figure 3A). 

Next, we investigated changes in the expression of the NO biosynthesis genes *NITRATE REDUCTASE* (*NR*) and *NITRITE REDUCTASE* (*NIR*) (Figure 3B). The expression of *NR* (g1372) and *NIR* (g59680), both genes involved in early NO biosynthesis, did not differ significantly between YJM and JM under control conditions. After flood treatment, their expression gradually increased in YJM but gradually decreased in JM. We also examined the expression of the NO-scavenging *PHYTOGLOBIN* (*PGB*) genes during early flooding treatment (Figure 3C). *PGB1* (g34086), *2* (g60867), and *3* (g21239) showed higher levels of expression in YJM than in JM under control conditions and during early flood treatment (0.5 days). Their expression decreased, however, in YJM by 3 days after flooding so that, by this timepoint, the expression of *PGB*s was higher in JM than in YJM.

### 3.4. The Responses of MDHAR during Early Flooding Treatment

Previous studies reported that ascorbic acid (AsA) and PGB are involved in NO metabolism in plants. In vitro enzymatic assays have shown that NO scavenging is catalyzed by monodehydroascorbic acid reductase (MDHAR), which mediates binding reactions involving iron reduction in PGB in the presence of AsA and NADH [28]. AsA supports NO scavenging, directly reducing PGB in plants to produce nitrate and monodehydroascorbate (MDA). The final product of this reaction is MDA, which is efficiently reduced back to AsA in the presence of MDHAR and NADH. NO scavenging is also promoted by MDHAR [29]. 

We therefore investigated *MDHAR* gene expression and MDHAR-associated enzymatic activity (Figure 4). It was confirmed that the expression of different *MDHAR* genes (cytosolic, g25138; peroximal, g17489; and mitochondrial, g35177) and MDHAR enzymatic activity were higher in the flood-tolerant YJM than in the flood-sensitive JM. Changes in MDHAR expression and activity in tolerant sweet potato during flood treatment may therefore play an important role in NO regulation through interactions with PGB as well as via ROS regulation.

## 4. Discussion

Plants that exhibit resistance to flooding use two general survival strategies: low-O_2_ escape syndrome (LOES) and low-O_2_ quiescence syndrome (LOQS) [2,30]. Regulated anaerobic metabolism in both LOES and LOQS is equivalent to survival in low-O_2_ conditions. At the heart of this mechanism is an evolutionarily conserved group of TFs, ethylene response factor VIIs (ERFVIIs), which activate the genes required for anaerobic metabolism in Arabidopsis [31]. A reduction in O_2_ levels in plants stabilizes ERFVII TFs such as RAP2.12 and RAP2.3, which play an important role in activating anaerobic responses at the transcriptional level [32]. Giuntoli et al. [33] also reported that the expression of a set of genes involved in the oxidative stress response is induced in flood-exposed Arabidopsis. Thus, ERFVII is a positive regulator of oxidative stress-related genes and genes involved in fermentative metabolism. The *ERFVII* TFs *RAP2.2*, *RAP2.3*, and *RAP2.12* mediate responses to oxidative stress and may act redundantly [34]. Overexpression of *RAP*-type *ERFVII* TFs also confers resistance to oxidative stress after H_2_O_2_ application. *RAP*-type *ERFVII* genes, which are involved in ROS scavenging and signaling, are themselves positively regulated by ERFVII TFs [33,34]. 

We investigated the expression of the representative *ERFVII* TFs *RAP2.3* and *RAP 2.12* in the sweet potato cultivars YJM and JM, which were identified as flood-resistant and flood-sensitive, respectively, in a previous study [21] (Figure 1). The expression of *ERF72/RAP2.3* and *ERF74/RAP2.12* was higher in YJM than in JM during flood stress; expression levels of these genes were increased, especially during the early period of flooding. We also measured the expression of four genes belonging to other groups within the ERF transcription factor family, *ERF1* (g31279), *ERF2* (g25395), *ERF4* (g54463), and *ERF5* (g20475). These showed either similar increases in expression in the two cultivars after flood treatment or slightly higher expression levels in JM (data not shown). A recent transcriptomic analysis of the sweet potato cultivars YJM and JM revealed changes in the expression of *ETR*, *EIN*, and *ERF*, which are all genes related to signal transduction involved in ET signaling pathways [21]. As indicated by previous studies in Arabidopsis and rice, it is highly likely that *RAP2.3* and *RAP2.12*, which are ERFVII TFs, are an important part of the mechanism enabling flood resistance in sweet potato.

The onset of anaerobic conditions triggers a burst of ROS in Arabidopsis due to changes in NADPH oxidase activity in membranes and an imbalance in the mitochondrial electron transfer system [35]. Arabidopsis *RBOHD* mutants are less tolerant of anaerobic conditions and show negative effects on *ALCOHOL DEHYDROGENASE1* (*ADH1*) expression, compared to wild-type seedlings immersed in water [35,36,37]. This suggests that ROS generated by *RBOHD* under these conditions may be a positive signal necessary for plant tolerance of flooding-mediated hypoxia. Yamauchi et al. [38] observed high levels of induction of *RBOH* expression in parallel with repression of *MT*, which encodes an ROS scavenger, in maize roots that did not show tissue reduction following treatment with the NADPH oxidase inhibitor diphenyleneiodonium (DPI). High expression of *RBOH* can therefore induce cell death in plants during flood treatment through oxidative burst. High *RBOH* expression appears to act either by suppressing the expression of ROS-scavenging genes, including the *MT*s, or, during flooding, through signal transduction pathways that increase the expression of a downstream gene that causes ROS to be removed and activates an appropriate defense mechanism. We observed that the expression of *RBOHA*, *D*, and *E* increased during the initial flood treatment in the flood-resistant cultivar YJM, and the expression of *MT2* significantly increased in the flooding treatment after 12 h (Figure 2). Therefore, flood tolerance in sweet potato appears to involve activation of the pathway involving the ROS scavenger protein MT.

Like ROS, NO is detrimental to plant cells, but it is also a key component of plant response-related signaling pathways [39]. In plants, NO is involved in the degradation of transcriptional regulators, which leads to the activation of key hypoxia-responsive genes [40]. Indeed, in anaerobic plants, NO availability negatively modulates the activation of responses induced by ERFVII TFs. Arabidopsis NR mutants such as *nitrate reductase 1* (*nia1*) and *nia2* exhibit changes in anaerobic gene transcription due to the impaired production of NO, which alters the activation of genes downstream of *ERFVII* in reaction to anaerobic conditions [40]. Therefore, NO destabilizes *ERFVII* and inhibits downstream signaling pathways. Hartman et al. [41] reported that early ET capture from flooded Arabidopsis increases transcription of the NO scavenger nonsymbiotic *PGB1*, thereby reducing the solubility of NO and promoting the stability of ERFVII. This phenomenon occurs prior to severe hypoxia in plants and acts as a priming event, enhancing the plant’s tolerance to upcoming stressful conditions. In the current study, a greater increase in NO_2_^−^ levels during flood treatment was observed in the flood-resistant cultivar YJM than in JM, a flood-sensitive cultivar (Figure 3A); although there was no significant change during the initial flood treatment, a difference was observed after 3 days of treatment. During the flood treatment, the expression of NR and NIR, genes involved in NO_2_^−^ levels, gradually increased in YJM but decreased gradually in JM (Figure 3B). It therefore appeared that exposure to flooding activated response mechanisms regulated by NO_2_^−^ generation in YJM, a flood-resistant sweet potato cultivar.

PGB, which scavenges NO, potentially promotes the interaction between NO and ET. Levels of *PGB* mRNA increase during submergence and hypoxic conditions in several plant species [42,43]. NO also increases the levels of *PGB* mRNA in rice, cotton, and spinach, indicating a feedback mechanism [44,45,46]. Interactions and feedback regulation between NO and ET have also been shown at the level of ERFVII, as ERFVII stability and action are highly dependent on NO and oxygen levels [40,47]. When NO or oxygen concentrations are reduced, ERFVII TFs accumulate, thereby promoting transcription of downstream target genes. Interestingly, several genes regulating ERFVII TFs contain hypoxia-responsive promoter elements, including several ET signaling genes and the NO-scavenging gene *PGB* [31]. Therefore, ERFVII TFs are upregulated by NO, oxygen, and ET, whereas ERFVII action can, in turn, induce ET biosynthesis and NO scavenging, creating a positive feedback loop [31]. We observed that the expression of *PGB*s increased strongly within 0.5 days of flood treatment in the flood-resistant cultivar YJM, i.e., during the initial period of flooding (Figure 3); in contrast, the expression of *PGB*s increased slowly and gradually in JM. In this study, therefore, the expression of NO_2_^−^ generation-related genes gradually increased in response to flooding, but the expression of *PGBs* increased immediately, suggesting that the overall level of NO_2_^−^ increased gradually in the flooding-tolerant YJM. It is possible that NO_2_^−^ generated through the nonenzymatic pathway via mitochondria may have influenced the increase in NO_2_^−^ levels, in addition to NO_2_^−^ generated by the enzymatic pathway acting through NR and NIR. These results suggest that increases in NO_2_^−^ generation and elimination affected the ROS signaling mechanism through the expression of *ERFVII*, thereby activating the flood resistance mechanism.

We previously reported comparative transcriptome profiling to compare a flood-tolerant sweet potato cultivar with a flood-sensitive cultivar [21]. A higher number of some of the ROS signaling-related genes such as mitogen-activated protein kinase (MAPK) and ET signaling-related genes were upregulated in the tolerant cultivars than in the susceptible cultivar. Recently, another study also reported comparative transcriptome profiling using different soybean cultivars [48]. A higher number of some of the ROS-related genes such as glutathione S-transferase and lipoxygenase were upregulated in the tolerant cultivars than in the susceptible cultivar. The number of some phytohormone ABA-related TFs of the basic leucine zipper domain was also higher in the tolerant cultivars than in the susceptible cultivar. Similar to our previous study, the expression levels of several candidate genes-related to ROS and phytohormones thought to be involved in flooding tolerance correlated with the comparative transcriptomic data.

This study addressed the transmission of the plant signaling molecules ET, ROS, and NO and showed how these signals were correlated in flood-tolerant and flood-sensitive cultivars of sweet potato (Figure 5). Consistent with previous research, a signaling pathway acting through ERFVII was activated in flood-resistant plants. In addition, our results suggest that the expression of *RBOH* and *MT* genes, which encode components of an ROS signaling pathway involving ERFVII, played an important role in enabling the flood resistance of sweet potato. This study also suggests that activation of the NO biosynthesis and scavenging cycle and an increase in MDHAR activity likely regulated ERFVII expression and levels of ROS via other pathways.

## 5. Conclusions

In conclusion, we characterized changes in the expression of flooding response-related genes in flood-tolerant and flood-sensitive sweet potato cultivars that were associated with response mechanisms mediated by ET, ROS, and NO. This approach identified candidate sweet potato genes whose expression was associated with changes in specific responses involved in metabolic and signaling pathways related to ET, ROS, and NO. The use of marker-assisted selection to identify genes linked to flood tolerance offers several advantages over flooding control in an integrated management system. Further investigation, based on large-scale genomic and transcriptomic analysis, is required to elucidate the exact role played by each candidate gene in regulating the signaling pathways involved in flooding tolerance responses of sweet potato during flooding stress. Transgenic plants with an enhanced or reduced expression of candidate genes will be generated to determine their role in the mechanisms conferring flooding tolerance. Overall, our results provide valuable information for enabling the development of crops with enhanced tolerance to hypoxic stress induced by flooding.

## Figures and Tables

**Figure 1 antioxidants-11-00878-f001:**
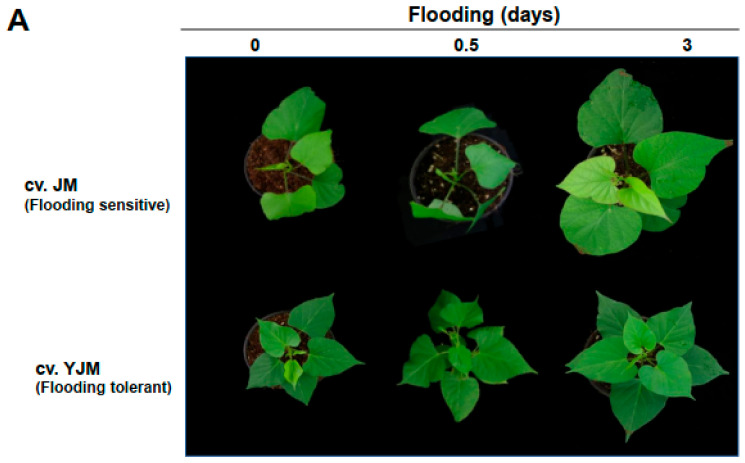
Effects of flooding stress on the sweet potato cultivars Yeonjami (YJM) and Jeonmi (JM). (**A**) Phenotypes of aboveground portions of YJM and JM after 3 d of flooding treatment. (**B**) Expression of *ERFVII* transcription factors in each cultivar after 0, 0.5, or 3 d of flooding treatment. Different letters represent statistically significant differences between control and flooding treatment, and between flooding-treated JM and YJM, determined using two-way ANOVA with the LSD post hoc test; *p* < 0.05.

**Figure 2 antioxidants-11-00878-f002:**
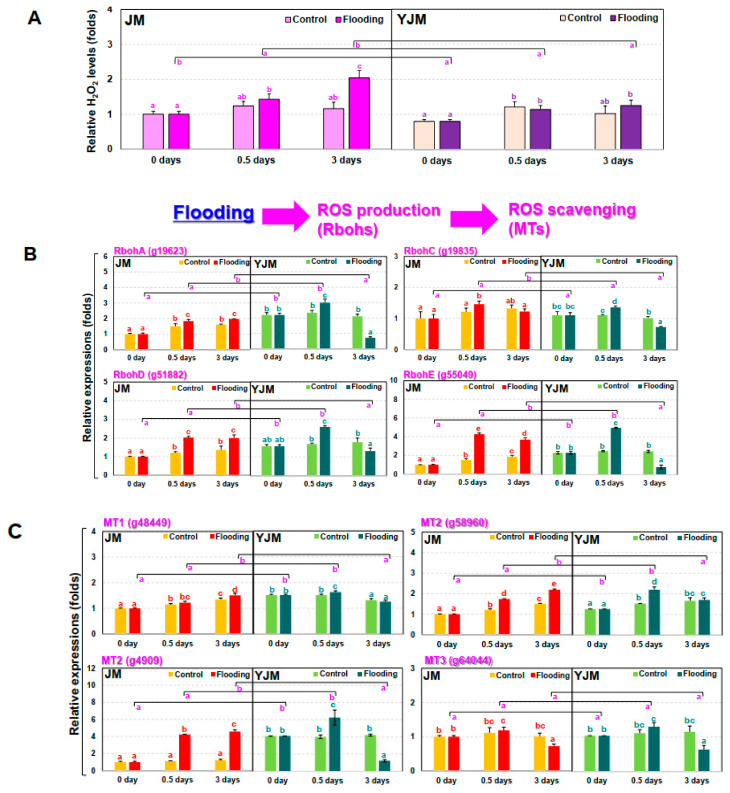
Expression of genes related to hydrogen peroxide (H_2_O_2_) and ROS pathways in sweet potato cultivars treated with flooding stress for 3 d. (**A**) Relative levels of H_2_O_2_ in each cultivar after 0, 0.5, or 3 d of flooding treatment. (**B**) Expression of *RBOH* genes in each cultivar after 0, 0.5, or 3 d of flooding treatment. (**C**) Expression of *MT* genes in each cultivar after 0, 0.5, or 3 d of flooding treatment. Different letters represent significant differences between control and flooding treatment, and between flooding-treated JM and YJM, determined using two-way ANOVA with the LSD post hoc test; *p* < 0.05.

**Figure 3 antioxidants-11-00878-f003:**
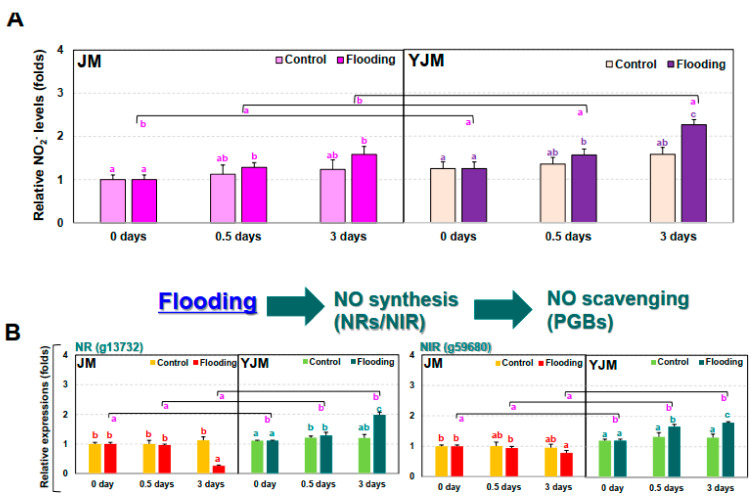
Levels of nitrite (NO_2_^−^) and expression of NO-related genes in sweet potato cultivars treated with flooding stress for 3 d. (**A**) Relative levels of NO_2_^−^ in each cultivar after 0, 0.5, or 3 d of flooding treatment. (**B**) Expression of the NO biosynthesis genes *NR* and *NIR* in each cultivar after 0, 0.5, or 3 d of flooding treatment. (**C**) Expression of *PGB* genes in each cultivar after 0, 0.5, or 3 d of flooding treatment. Different letters represent statistically significant differences between control and flooding treatment, and between flooding-treated JM and YJM, determined using two-way ANOVA with the LSD post hoc test; *p* < 0.05.

**Figure 4 antioxidants-11-00878-f004:**
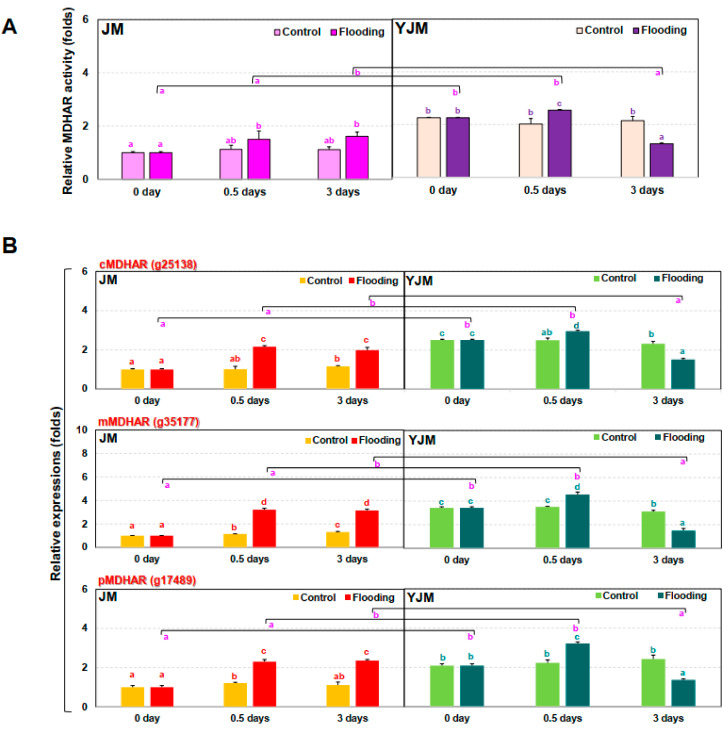
Changes in MDHAR activity and *MDHAR* gene expression in sweet potato cultivars treated with flooding stress for 3 d. (**A**) Relative MDHAR activity in each cultivar after 0, 0.5, or 3 d of flooding treatment. (**B**) Expression of *MDHAR* genes in each cultivar after 0, 0.5, or 3 d of flooding treatment. c*MDHAR*: cytosolic *MDHAR*; p*MDHAR*: peroxisomal *MDHAR*; m*MDHAR*: mitochondrial *MDHAR*. Different letters represent statistically significant differences between control and flooding treatment, and between flooding-treated JM and YJM, determined using two-way ANOVA with the LSD post hoc test; *p* < 0.05.

**Figure 5 antioxidants-11-00878-f005:**
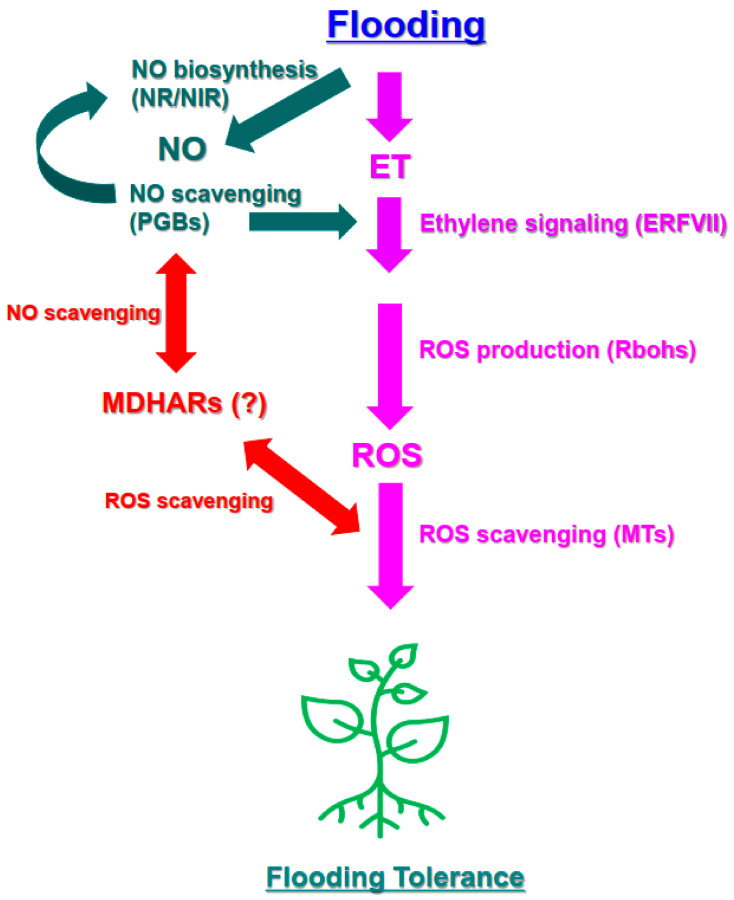
A suggested model of the ET-, ROS-, and NO-related biological processes and genes involved in influencing the flooding response that occurs in sweet potato leaves under early flooding. Flooding conditions result in the induction of ERFVII transcription factors and the expression of RBOHs, which produce O_2_^−^ radicals from oxygen in the apoplast. RBOH expression may enhance RBOH activity and increase ROS levels in the apoplast as well as the cytosol. Expression of genes encoding ROS scavengers, such as MTs, increases, leading to an increase in ROS signaling and scavenging responses and inducing the response in flooding-tolerant sweet potato. The NO synthesis-scavenging cycle regulates ERFVII-mediated signaling. MDHARs are also potentially involved in NO/ROS-mediated responses in sweet potato under early flooding conditions. ERF: ethylene response factor; ET: ethylene; ROS: reactive oxygen species; RBOH: respiratory burst oxidase homolog; MT: metallothionein; NO: nitric oxide; NR: nitrate reductase; NIR: nitrite reductase; PGB: phytoglobulin; MDHAR: monodehydroascorbate reductase.

## Data Availability

Data are contained within the article.

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
