# Peer review of "Flooding Tolerance in Sweet Potato (Ipomoea batatas (L.) Lam) Is Mediated by Reactive Oxygen Species and Nitric Oxide"

_antioxidants, 2022, doi:10.3390/antiox11050878_

Round 1
Reviewer 1 Report
Review Comments
The topic addressed is interesting but not original, in fact many indications are already found in other publications. The manuscript has its merit, but it needs a careful review before its publication. Thus, my suggestion is acceptance after some minor revisions and successful addressing of comments.
Abstract
The abstract does not adequately describe the study nor does it arouse interest in readers to read the manuscript. The authors must always remember the basics of the what was studied, how was the study conducted, what were the key findings, what are the implications of the results.
Key words
You cannot use the same words that are in the title (nitric oxide, reactive oxygen species, sweet potato); the main purpose of having key words is to increase chances of MS getting a hit during web search.
Introduction
Even though the introduction provides background on various interactions during flood stress, I feel some sentences should be improved to make it more readable and engaging. Plus, the authors must clearly say about the study objectives. Also, it will be good if the authors can justify the selection of sweet potato for this study, although they had done some previous studies.
Lines 84-85, The spatiotemporal dynamics of NO and its effects may vary therefore depending on the flood conditions…..Is this statement is just authors assumption or reported?
Material and methods
Under what soil conditions was the sweet potato plants were grown?
Why don’t you study the changes in antioxidant enzymes activities under flood stress?
Results
Almost well presented.
Discussion
This section seems to be ok but not perfect. What missing is the lack of discussion of the results with recent research works. The authors must discuss their results with recent works.
Conclusion
The authors have made a good conclusion of their work.
I strongly recommend the authors to include an Abbreviation section since many abbreviations are used in this manuscript.
Author Response
The topic addressed is interesting but not original, in fact many indications are already found in other publications. The manuscript has its merit, but it needs a careful review before its publication. Thus, my suggestion is acceptance after some minor revisions and successful addressing of comments.
Abstract
The abstract does not adequately describe the study nor does it arouse interest in readers to read the manuscript. The authors must always remember the basics of the what was studied, how was the study conducted, what were the key findings, what are the implications of the results.
â–¶As your comments, we edited the sentences in the ‘Abstract’ part of revised manuscripts (Line 7 and 16 of page 2).
Key words
You cannot use the same words that are in the title (nitric oxide, reactive oxygen species, sweet potato); the main purpose of having key words is to increase chances of MS getting a hit during web search.
â–¶As your comments, we edited the key wards in the revised manuscripts (Line 22 of page 2).
Introduction
Even though the introduction provides background on various interactions during flood stress, I feel some sentences should be improved to make it more readable and engaging. Plus, the authors must clearly say about the study objectives. Also, it will be good if the authors can justify the selection of sweet potato for this study, although they had done some previous studies.
â–¶As your comments, we edited the sentences in the ‘Introduction’ part of revised manuscripts (Line 1 of page 6).
Lines 84-85, The spatiotemporal dynamics of NO and its effects may vary therefore depending on the flood conditions…..Is this statement is just authors assumption or reported?
â–¶ It is report. As your comments, we added the references 15, 16 and 17 in the ‘Introduction’ part of revised manuscripts (Line 14 of page 5).
Material and methods
Under what soil conditions was the sweet potato plants were grown?
â–¶ As your comments, we described about cultivation conditions in the ‘Materials and methods’ part (Materials and methods section 2.1.) of revised manuscripts (Line 13 of page 6).
Why don’t you study the changes in antioxidant enzymes activities under flood stress?
â–¶ The purpose of this study is to study the influence of ethylene, ROS and NO in sweetpotato during flooding treatment. In studies on the control mechanism of ROS in plants during flood treatment, the Rboh gene involved in the ROS production and the MT gene involved in the ROS remove have been reported in many studies. Thus, the expression change of Rbohs and MTs was confirmed with the gene. In addition, there has been a report on the function of MDHAR in relation to the mechanism of NO regulation, and thereby the change in the activity and expression of MDHAR was investigated.
Results
Almost well presented.
â–¶ Thanks for your comments.
Discussion
This section seems to be ok but not perfect. What missing is the lack of discussion of the results with recent research works. The authors must discuss their results with recent works.
â–¶As your comments, we added the sentences about recently study in the ‘Discussion’ part of revised manuscripts (Line 23 of page 14).
Conclusion
The authors have made a good conclusion of their work.
â–¶ Thanks for your comments.
I strongly recommend the authors to include an Abbreviation section since many abbreviations are used in this manuscript.
â–¶ As your comments, we added the abbreviations in the revised manuscripts (Line 25 of page 2).
Reviewer 2 Report
This manuscript presents a follow-up study of an earlier transcriptomic analysis of sweetpotato leaves under flooding. It contains some more qPCR analyses as well as some metabolite contents.
Although the plant system and its sensitivity toward flooding is of relevance for the scientific community, this current study has many severe issues and errors that require extensive modifications before publication. I am only listing the most severe problems here.
First of all, from this and the earlier study it does not become clear which leaves were actually harvested and analyzed. You indicated that you flooded about 65% of the aboveground leaves. Did you harvest the leaves below the water surface, or the leaves above? Were the leaves in illumination at time of stress and harvest? This strongly influences the oxygen content within tissues. If illumination was present, there is no oxygen deficiency due to ongoing photosynthesis within the first hours. At which time of day, relative to start of illumination, was the flooding treatment started?
As far as I understand your system, and based on your expression data in your earlier paper, you do not have hypoxia at all in your system within the observed time frame. Therefore, the statement in line 363 and possibly other parts of the manuscript is wrong.
Second, the qPCR data from this current study completely contradict the results from the transcriptome study. I checked all expression values. In this study, many of the tested genes were down-regulated by flooding in the tolerant cultivar after 3 h, while they were up-regulated in the transcriptome dataset.
Furthermore, there is one mis-labelling, ERF74 is not g60549, but g60949. Did you use correct gene primers?
Also, statistics for the qPCR were not correctly done. You should do two-way ANOVA and also test whether differences between timepoints in each cultivar exist. You did not at all test for the stress effect, and therefore you cannot make any statement on gene induction or repression right now (i.e., line 322ff).
Third, it is not correct that ethylene controls the major metabolic responses under flooding. ERFVIIs are influenced at the transcriptional level by ethylene, but the major regulation is at the post-translational level through oxygen and NO. Line 53ff is therefore wrong, as well as lines 159ff. And not all ERFs are ethylene-regulated after all.
Did you actually check expression of the hypoxia-inducible ERFVIIs, HRE1 and HRE2?
ROS are not only produced by RBOHs under hypoxia (RBOHs are rather involved in ROS production under pathogen attack) -> line 63ff. Under hypoxia, ROS might be formed by mitochondria and/or the photosynthetic electron transport chain.
Line 80, what do you mean with "atmospheric plant tissue"?
Analysis of NO content (line 118ff) -> this technique is completely wrong. The Griess reagent is for detection of nitrite, not NO. Furthermore, you cite a paper (Tossi et al.) that does not even use this reagent. Moreover, that paper was retracted and should not be cited anymore.
Figure 1, I actually cannot see any differences between the plants. In my opinion YJM looks even a bit more stressed.
In general, you over-interpret the data. You present gene expression data and some molecules, but you have no evidence for their interaction or influence between them, as you imply in Fig. 5 and many parts of the text. The only thing you can assume is a correlation, but no causal relationship.
There are many small errors in the text. For example, a gene is not involved in a pathway, only the gene product/ protein. RBOH is not involved in ROS signaling but in ROS production.
The discussion is very long and is partly not related to the data of this manuscript, for example the section on LOES/LOQS. You do not do any analysis on this topic, and you even do not mention which strategy sweetpotato would follow if it would be fully submerged. Also, the section on ROS and aerenchyma is not needed since you analyzed leaves, which probably do not even form aerenchyma.
Line 262, you mean ref 21, not 20 here?
Author Response
Dear Editor and Reviewers of Antioxidants,
I would like to send you the revised manuscript for the possible publication in Antioxidants. I did our best to revise the manuscript according to the valuable comments of you and reviewers. I carefully revised the manuscript according to Antioxidants CHECKLIST/COMMENTS. The revised sentences were written by red letter in the manuscript. The answers to the valuable comments of you and reviewers are as follows;
Comments and Suggestions for Authors
Review Comments-2
This manuscript presents a follow-up study of an earlier transcriptomic analysis of sweetpotato leaves under flooding. It contains some more qPCR analyses as well as some metabolite contents. Although the plant system and its sensitivity toward flooding is of relevance for the scientific community, this current study has many severe issues and errors that require extensive modifications before publication. I am only listing the most severe problems here.
1.1. First of all, from this and the earlier study it does not become clear which leaves were actually harvested and analyzed. You indicated that you flooded about 65% of the aboveground leaves. Did you harvest the leaves below the water surface, or the leaves above? Were the leaves in illumination at time of stress and harvest? This strongly influences the oxygen content within tissues. If illumination was present, there is no oxygen deficiency due to ongoing photosynthesis within the first hours. At which time of day, relative to start of illumination, was the flooding treatment started?
â–¶ As your comments, we added the following sentences in the ‘Results’ part (Results section 3.1.) of revised manuscripts (Line 16 of page 8)
“Approximately 65% of the aboveground tissue of 2-month-old plants was submerged by adding excess water to the pot in a growth chamber at 25℃ under a 16 h/8 h light/dark photocycle. When the susceptible cultivar JM and the resistant cultivar YJM were exposed to flooding for 0, 0.5, and 3 days, the above atmospheric leaves of the JM plants showed slight wilting and curling, whereas the leaves of the YJM plants were less damaged (Fig. 1A).”
1.2. As far as I understand your system, and based on your expression data in your earlier paper, you do not have hypoxia at all in your system within the observed time frame. Therefore, the statement in line 363 and possibly other parts of the manuscript is wrong.
â–¶ As your comments, we removed the ‘hypoxia’ in the revised manuscripts including legend of Figure 5 (Line 10 of page 24).
2.1. Second, the qPCR data from this current study completely contradict the results from the transcriptome study. I checked all expression values. In this study, many of the tested genes were down-regulated by flooding in the tolerant cultivar after 3 h, while they were up regulated in the transcriptome dataset.
â–¶ The reasons for the difference between the qPCR results of this study and the transcriptome data of previous studies are as follows:
- This study was independently conducted under the same conditions as previous studies. Therefore, the qPCR results may be similar to, but not consistent with, the DEG results of previous studies.
- We found serious errors in the supplemental data of previous studies published in the journal. The DEGs results of JM and YJM were stored in reverse in the supplemental data. Therefore, the DEGs of JM in the supplemental data is the DEGs of YJM, and the DEGs of YJM result is the DEGs of JM. However, the results of figures including heatmap and qPCR used in the previous paper were used correctly. Therefore, a similar pattern can be identified by reviewers if the transcriptome results of supplemental data from the previous study are reversed and compared with the qPCR results of this study.
We are currently in the process of revising the supplemental data of previously published our paper to the journal. Therefore, the qPCR results of this study are considered to be no problem.
2.2. Furthermore, there is one mis-labelling, ERF74 is not g60549, but g60949. Did you use correct gene primers?
â–¶ You are correct. It's our mistake. New experimental results of g60949 are added to Figure 1B.
2.3. Also, statistics for the qPCR were not correctly done. You should do two-way ANOVA and also test whether differences between timepoints in each cultivar exist. You did not at all test for the stress effect, and therefore you cannot make any statement on gene induction or repression right now (i.e., line 322ff).
â–¶ As your comments, the control group was added for each time period in all data, and was marked using two-way ANOVA.
3.1. Third, it is not correct that ethylene controls the major metabolic responses under flooding. ERFVIIs are influenced at the transcriptional level by ethylene, but the major regulation is at the post-translational level through oxygen and NO. Line 53ff is therefore wrong, as well as lines 159ff. And not all ERFs are ethylene-regulated after all.
â–¶ As your comments, we added the following sentences in the ‘Results’ part (Results section 3.1.) of revised manuscripts (Line 23 of page 8)
“TFs belonging to the ERFVII group, including ERF71/HER2, ERF72/RAP2.3, ERF74/RAP2.12, and ERF75/RAP2.2, are important representatives of genes that function in the ET signaling pathway activated during flood treatment in the transcription and post-translational levels [27]. Although expression analysis at the post-translational level is more important in the ERFVIIs, in this study, expression of ERFVII group genes was investigated during early flood treatment at the transcription level (Fig. 1B)”
3.2. Did you actually check expression of the hypoxia-inducible ERFVIIs, HRE1 and HRE2?
â–¶ Unfortunately, HER1 and HRE2 were not found in the sweetpotato transcriptome database.
3.3. ROS are not only produced by RBOHs under hypoxia (RBOHs are rather involved in ROS production under pathogen attack) -> line 63ff. Under hypoxia, ROS might be formed by mitochondria and/or the photosynthetic electron transport chain.
â–¶ As your comments, we added the following sentences in the ‘Introduction’ part of revised manuscripts (Line 3 of page 5)
“ROS also might be formed by mitochondria and/or the photosynthetic electron transport chain [8,10].”
- Line 80, what do you mean with "atmospheric plant tissue"?
â–¶ It means the not submerged atmospheric plant tissues. We added the sentences in the in the ‘Introduction’ part of revised manuscripts (Line 14 of page 5).
- Analysis of NO content (line 118ff) -> this technique is completely wrong. The Griess reagent is for detection of nitrite, not NO. Furthermore, you cite a paper (Tossi et al.) that does not even use this reagent. Moreover, that paper was retracted and should not be cited anymore.
â–¶ As your comments, ‘nitric oxide’ change to ‘nitrite’ and incorrect references were removed in the ‘Materials and methods’ and ‘Results’ parts (Materials and methods section 2.3. and Results section 3.3.) of revised manuscripts (Line 9 of page 7 and line 17 of page 10) and Figure 3.
- Figure 1, I actually cannot see any differences between the plants. In my opinion YJM looks even a bit more stressed.
â–¶ The cultivar names are mislabeled in the photo of Figure 1A. Corrected cultivar names again.
- In general, you over-interpret the data. You present gene expression data and some molecules, but you have no evidence for their interaction or influence between them, as you imply in Fig. 5 and many parts of the text. The only thing you can assume is a correlation, but no causal relationship.
â–¶As your comments, we changed the following sentences in the legends of Figure 5 (Line 12 of page 24).
“A suggested model of the ET, ROS, and NO-related biological processes and genes involved influence in the flooding response that occurs in sweetpotato leaves under early flooding.”
- There are many small errors in the text. For example, a gene is not involved in a pathway, only the gene product/ protein. RBOH is not involved in ROS signaling but in ROS production.
â–¶As your comments, we changed the sentences in the ‘Results’ part (Results section 3.3.) of revised manuscripts (Line 19 of page 9) and Figure 2.
- The discussion is very long and is partly not related to the data of this manuscript, for example the section on LOES/LOQS. You do not do any analysis on this topic, and you even do not mention which strategy sweetpotato would follow if it would be fully submerged. Also, the section on ROS and aerenchyma is not needed since you analyzed leaves, which probably do not even form aerenchyma.
â–¶As your comments, we changed the sentences in the ‘Discussion’ part of revised manuscripts (Line 4 of page 11 and line 5 of page 12).
- Line 262, you mean ref 21, not 20 here?
â–¶ You are correct. It's our mistake. The reference [20] change to [21] in the discussion part of revised manuscripts (Line 2 of page 12).
Reviewer 3 Report
The paper is well written and is interesting, however are necessary some changes and integrations.
Title: add also the initial of the author name of the species studied: L. if is Linneo or....). Also in the paragraph 2.1
Line 110 Specify how many and which leaves of the plants were collected for the analyses. The chose was random?
Section 2.2 Describe better the colorimetric method utilized. Specify better the sentence "H2O2 measurements were expressed as relative values". Please make explicit the term "relative value"
Fig 1:the leaves of the JM plants showed slight wilting and
curling, whereas the leaves of the YJM plants were less damaged.
Watching the picture it is not observed wilting and curling and damnages in cv. JM sensitive!
In the Discussion Section are presented of the results with the figure 4! Why?
In the discussion there are phrases what are reported also in the results section. Eliminate it as already presented as results.
Author Response
Dear Editor and Reviewers of Antioxidants,
I would like to send you the revised manuscript for the possible publication in Antioxidants. I did our best to revise the manuscript according to the valuable comments of you and reviewers. I carefully revised the manuscript according to Antioxidants CHECKLIST/COMMENTS. The revised sentences were written by red letter in the manuscript. The answers to the valuable comments of you and reviewers are as follows;
Comments and Suggestions for Authors
Review Comments-3
The paper is well written and is interesting, however are necessary some changes and integrations.
- Title: add also the initial of the author name of the species studied: L. if is Linneo or....). Also in the paragraph 2.1
â–¶ As your comments, we changed to the ‘[Ipomoea batatas (L.) Lam]’ in the ‘Title’ and ‘Materials and methods’ parts (Materials and methods section 2.1.) of revised manuscripts (Line 1 of page 1 and line 9 of page 6)
- Line 110 Specify how many and which leaves of the plants were collected for the analyses. The chose was random?
â–¶ As your comments, we added the following sentences in the ‘Materials and methods’ part (Materials and methods section 2.1.) of revised manuscripts (Line 18 of page 6)
“For leaf samples used in the experiment, the 3rd and 4th leaves from the top were used in four plants, respectively.”
- Section 2.2 Describe better the colorimetric method utilized. Specify better the sentence "H2O2 measurements were expressed as relative values". Please make explicit the term "relative value"
â–¶ As your comments, we added the following sentences in the ‘Materials and methods’ part (Materials and methods section 2.2.) of revised manuscripts (Line 24 of page 6)
“A total of 0.1 g of leaf tissue was ground and homogenized in a solution of 50 mM potassium phosphate buffer (pH 6.5) and 1 mM hydroxylamine. The homogenate was centrifuged at 12,000 g for 15 min at 4 °C. A total of 100 μL of the supernatant was added to a reaction mixture containing ferric ammonium sulfate (FeNH4[SO4]), 25 mM sulphuric acid, 250 μM xylenol orange, and 100 mM sorbitol. After 30 min under the dark at room temperature for incubation, the absorbance of the samples was determined at 560 nm. H2O2 measurements were expressed as relative values by control and treatment.”
- Fig 1:the leaves of the JM plants showed slight wilting and curling, whereas the leaves of the YJM plants were less damaged. Watching the picture it is not observed wilting and curling and damnages in cv. JM sensitive!
â–¶ The cultivar names are mislabeled in the photo of Figure 1A. Corrected cultivar names again.
- In the Discussion Section are presented of the results with the figure 4! Why?
â–¶ Since the description of Figure 4 are based on the contents explained in the 'Discussion' section, they must be described together with the contents of the 'Discussion' section, so they are included in the 'Discussion' section, not in the 'Results' section.
- In the discussion there are phrases what are reported also in the results section. Eliminate it as already presented as results.
â–¶As your comments, we removed the sentences including results section.
Round 2
Reviewer 2 Report
In their revision, the authors modified their manuscript, but in my opinion not yet satisfactory. Furthermore, they raised some severe problems in their previous publication, so that one can only fully judge this follow-up study once the original article was adequately corrected.
Besides this issue, there are many other problems that need to be corrected by the authors.
First, the figures did not get better but rather worse in this revision. Please do not use the yellow and orange background in Fig. 2, 3 and 4. Please stick with one color scheme for the two genotypes and treatments, for example the one from Fig. 1B. Furthermore, please use letters instead of asterisks to show significant differences. The current way is very hard to follow.
Second, why do you not write a result section on Fig. 4? Those data suddenly appear in the discussion but should be described in the results as well.
Third, there are several problems with wording, related to either English style or errors on the content. I am only listing the most severe here, more can be present.
Line 23-24, this part of the sentence is hard to understand, and you are not working with defense after all, rather with acclimation responses. The term "defense" should be only used in biotic stress responses. See also lines 34 and 96, and many more places in the text and in Fig. 5.
Line 27, ERFVIIs are, to my knowledge, not involved in ethylene signaling, but in low-oxygen signaling. The big group of "ethylene response factors" is only partially involved in ethylene signaling, and many members have completely different roles.
Line 54, ethylene production is not a metabolic response, at least not related to primary metabolism.
Line 82, add "under waterlogging" after "cotton" to make this sentence useful.
Line 111, it is still not clear whether you harvested the leaves above or below the water surface. This information is absolutely required to judge the expression data.
Fig. 1A, the plants still look very much alike, and the difference in your previous publication was much stronger. Also, the leaf shape between both cultivars differed quite a bit in your previous work, which is not seen here. Do you have other pictures to show?
Line 176, give a reference to this statement.
Line 168, figure legend contains errors.
Line 212, what is the meaning of the word "text" here?
Line 222, do not use the word "confirmed" since the studies you mentioned measured NO and not nitrite. Use rather the word "tested". Can you indicate or justify why you measured nitrite, but no NO levels?
Line 249, ref 26 is not the original reference that demonstrates this.
Line 249, reduction in oxygen levels STABILIZE ERFVIIs.
Line 256, which genes do you mean by "these genes"?
Line 270, ricetca?
Line 275, rbohd mutants are not more tolerant to hypoxia, but less tolerant.
Line 289-290, this is only true for the first 12 h, not for the 3d-time point.
Line 293, re-phrase, since the second part of the sentence might be mis-leading.
Line 309-310, there is no justification for this statement in your dataset. You did not measure NO levels. Same is true for line 327.
Line 333ff, hemoglobins have been re-named to phytoglobins. Stick with the new term throughout the manuscript (you already did so in the other parts).
Line 352, error in legend.
Line 371-373, this statement is also not justified.
Author Response
Dear Editor and Reviewer of Antioxidants,
I would like to send you the revised manuscript for the possible publication in Antioxidants. I did our best to revise the manuscript according to the valuable comments of you and reviewers. I carefully revised the manuscript according to Antioxidants CHECKLIST/COMMENTS. The revised sentences were written by red letter in the manuscript. The answers to the valuable comments of you and reviewers are as follows;
- First, the figures did not get better but rather worse in this revision. Please do not use the yellow and orange background in Fig. 2, 3 and 4. Please stick with one color scheme for the two genotypes and treatments, for example the one from Fig. 1B. Furthermore, please use letters instead of asterisks to show significant differences. The current way is very hard to follow.
â–¶ As your comments, we edited the Figures.
- Second, why do you not write a result section on Fig. 4? Those data suddenly appear in the discussion but should be described in the results as well.
â–¶ As your comments, we edited the result section on Figure 4 in the ‘Results’ parts of revised manuscripts (Line 9 of page 11).
- Third, there are several problems with wording, related to either English style or errors on the content. I am only listing the most severe here, more can be present.
1) Line 23-24, this part of the sentence is hard to understand, and you are not working with defense after all, rather with acclimation responses. The term "defense" should be only used in biotic stress responses. See also lines 34 and 96, and many more places in the text and in Fig. 5.
â–¶ As your comments, we removed ‘defense’ in the revised manuscripts including Abstract section (Lines 7 and 18 of page 4, line 1 of page 6, line 1 of page 16, and line 10 of page 24).
2) Line 27, ERFVIIs are, to my knowledge, not involved in ethylene signaling, but in low-oxygen signaling. The big group of "ethylene response factors" is only partially involved in ethylene signaling, and many members have completely different roles.
â–¶ As your comments, we edited the sentences in the ‘Abstract’ part of revised manuscript (Line 12 of page 2).
3) Line 54, ethylene production is not a metabolic response, at least not related to primary metabolism.
â–¶ As your comments, we edited the following sentences in the ‘Introduction’ part of revised manuscript (Line 14 of page 4).
“One of the responses to flooding is the induction of ET [3]”.
4) Line 82, add "under waterlogging" after "cotton" to make this sentence useful.
â–¶ As your comments, we edited the sentences in the ‘Introduction’ part of revised manuscript (Line 14 of page 5).
5) Line 111, it is still not clear whether you harvested the leaves above or below the water surface. This information is absolutely required to judge the expression data.
â–¶ As your comments, we edited the following sentences in the ‘Materials and methods’ part of revised manuscript (Line 19 of page 6).
“The harvested leaves were above the water surface.”
6) Fig. 1A, the plants still look very much alike, and the difference in your previous publication was much stronger. Also, the leaf shape between both cultivars differed quite a bit in your previous work, which is not seen here. Do you have other pictures to show?
â–¶ As your comments, we changed the Figure 1A with other photos.
7) Line 176, give a reference to this statement.
â–¶ As your comments, we added new reference in the ‘Discussion’ and ‘References’ part of revised manuscript (Line 3 of page 12 and line 2 of page 20).
8) Line 168, figure legend contains errors.
â–¶ This is an error that occurs on the system while converting to the journal's PDF file. There are no errors in the original sentences. (Line 5 of page 23).
9) Line 212, what is the meaning of the word "text" here?
â–¶ Sorry, it is our mistake. We removed ‘text’ in the sentences in the ‘Results’ part of revised manuscript (Line 4 of page 10).
10) Line 222, do not use the word "confirmed" since the studies you mentioned measured NO and not nitrite. Use rather the word "tested". Can you indicate or justify why you measured nitrite, but no NO levels?
â–¶ As your comments, we edited the following sentences in the ‘Results’ part of revised manuscripts (Lines 15 and 21 of page 10).
“Plants usually regulate endogenous NO levels via the control of biosynthesis and scavenging. The main source of NO production is via enzymatic and nonenzymatic reduction of nitrite (NO2–) [13,14]. NO2– production is highly dependent on nitrate reductase (NR). The dependency of NO production on nitrite availability makes NR the major player in NO production [14,15]. The NO generation during hypoxia is also thought to result from enhanced NR activity and NO2– accumulation, providing a substrate for NO production [13,19].”
11) Line 249, ref 26 is not the original reference that demonstrates this.
â–¶ As your comments, we changed new reference in the ‘Results’ and ‘References’ part of revised manuscript (Line 3 of page 12 and line 11 of page 20).
12) Line 249, reduction in oxygen levels STABILIZE ERFVIIs.
â–¶ As your comments, we edited the sentences in the ‘Discussion’ part of revised manuscript (Line 4 of page 12).
13) Line 256, which genes do you mean by "these genes"?
â–¶ “these genes” means “RAP-type ERFVII”. The edited sentences are in the ‘Discussion’ part of revised manuscript (Line 10 of page 12).
14) Line 270, ricetca?
â–¶ Sorry, it is our mistake. We change ‘ricetca’ to ‘rice’ in the ‘Discussion’ part of revised manuscript (Line 24 of page 12).
15) Line 275, rbohd mutants are not more tolerant to hypoxia, but less tolerant.
â–¶ As your comments, we edited the sentences in the ‘Discussion’ part of revised manuscript (Line 3 of page 13).
16) Line 289-290, this is only true for the first 12 h, not for the 3d-time point.
â–¶ As your comments, we edited the sentences in the ‘Discussion’ part of revised manuscript (Line 16 of page 13).
17) Line 293, re-phrase, since the second part of the sentence might be mis-leading.
â–¶ As your comments, we edited the following sentences in the ‘Discussion’ part of revised manuscript (Line 19 of page 13).
“Like ROS, NO is detrimental to plant cells, but it is also a key component of plant response-related signaling pathways [39].”
18) Line 309-310, there is no justification for this statement in your dataset. You did not measure NO levels. Same is true for line 327.
â–¶ As your comments, we edited the following sentences in the ‘Discussion’ part of revised manuscripts (Lines 7 and 24 of page 14).
“During flood treatment, expression of NR and NIR, genes involved in NO2– levels, gradually increased in YJM, but decreased gradually in JM (Fig. 3B). It therefore appeared that exposure to flooding activated response mechanisms regulated by NO2– generation in YJM, a flood-resistant sweetpotato cultivar.”
“In this study, therefore, expression of NO2– generation-related genes gradually increased in response to flooding, but the expression of PGBs increased immediately, suggesting that the overall level of NO2– increased gradually increased in flooding tolerant YJM. It is possible that NO2– generated through the nonenzymatic pathway via mitochondria may have influenced the increase in NO2– levels, in addition to NO2– generated by the enzymatic pathway acting through NR and NIR. These results suggested that increases in NO2– generation and elimination affected the ROS signaling mechanism through the expression of ERFVII, thereby activating the flood resistance mechanism.”
19) Line 333ff, hemoglobins have been re-named to phytoglobins. Stick with the new term throughout the manuscript (you already did so in the other parts).
â–¶ As your comments, we edited the sentences in the ‘Results’ part of revised manuscript (Line 10 of page 11).
20) Line 352, error in legend.
â–¶ This is an error that occurs on the system while converting to a PDF file. There are no errors in the original manuscripts following sentences. This is an error on the system of converting the journal's PDF file (Line 5 of page 24).
21) Line 371-373, this statement is also not justified.
â–¶ As your comments, we edited the following sentences in the ‘Discussion’ part of revised manuscripts (Lines 22 of page 15).
“This study also suggested that activation of the NO biosynthesis and scavenging cycle and an increase in MDHAR activity likely regulated ERFVII expression and levels of ROS via other pathways.”
Reviewer 3 Report
The paper in this revised form can be published
Regards
Author Response
Thanks for your good comments.